# Effectiveness of a Psychoeducational Group Intervention Carried Out by Nurses for Patients with Depression and Physical Comorbidity in Primary Care: Randomized Clinical Trial

**DOI:** 10.3390/ijerph18062948

**Published:** 2021-03-13

**Authors:** Antonia Raya-Tena, María Isabel Fernández-San-Martin, Jaume Martin-Royo, Rocío Casañas, Glòria Sauch-Valmaña, Cèlia Cols-Sagarra, Elena Navas-Mendez, Roser Masa-Font, Marc Casajuana-Closas, Quintí Foguet-Boreu, Eva María Fernández-Linares, Jacobo Mendioroz-Peña, Susana González-Tejón, Luis Miguel Martín-López, María Francisca Jiménez-Herrera

**Affiliations:** 1Centre d’Atenció Primària Raval Nord, Institut Català de la Salut, 08001 Barcelona, Spain; 2Nursing Department, Faculty of Nursing, Rovira and Virgili University, 43002 Tarragona, Spain; maria.jimenez@urv.cat; 3Unitat de Suport a la Recerca Barcelona ciutat, Fundació Institut Universitari per a la recerca a l’Atenció Primària de Salut Jordi Gol i Gurina (IDIAPGol), 08001 Barcelona, Spain; mifsanmartin.bcn.ics@gencat.cat (M.I.F.-S.-M.); jmartinr.bcn.ics@gencat.cat (J.M.-R.); enavas@idiapjgol.info (E.N.-M.); 4Unitat Docent Multiprofesional Gerència Territorial Barcelona, Institut Català de la Salut, 08001 Barcelona, Spain; 5Unitat Básica de Prevenció, Gerència Territorial de Barcelona, Institut Català de la Salut, 08001 Barcelona, Spain; 6Research Departament, Associació Higiene Mental Les Corts, 08001 Barcelona, Spain; rocio.casanas@chmcorts.com; 7Unitat de Suport a la Recerca Catalunya Central, Fundació Institut Universitari per a la recerca a l’Atenció Primària de Salut Jordi Gol i Gurina (IDIAPGol), 08001 Barcelona, Spain; gsauch.cc.ics@gencat.cat (G.S.-V.); jmendioroz@gencat.cat (J.M.-P.); 8Centre d’Atenció Primària Martorell Rural, Institut Català de la Salut, 08001 Barcelona, Spain; celiacolssagarra@gmail.com; 9Centre d’Atenció Primària Besos, Institut Català de la Salut, 08001 Barcelona, Spain; rmasa.bcn.ics@gencat.cat; 10Fundació Institut Universitari per a la recerca a l’Atenció Primària de Salut Jordi Gol i Gurina (IDIAPGol), 08001 Barcelona, Spain; mcasajuana@idiapjgol.info; 11Department of Psychiatry, Vic University Hospital. Francesc Pla el Vigatà, 1, 08500 Vic, 08001 Barcelona, Spain; 42292qfb@comb.cat; 12Faculty of Medicine, University of Vic-Central University of Catalonia (UVic-UCC), 08500 Barcelona, Spain; 13Centre d’Atenció Primària Raval Sud, Línea Pediàtrica, Institut Català de la Salut, 08001 Barcelona, Spain; evafernandez.bcn.ics@gencat.cat; 14Health Promotion in Rural Areas Research Group (PRoSaARu), Gerència Territorial de la Catalunya Central, Catalan Health Institute, Sant Fruitós del Bages, 08001 Barcelona, Spain; 15Centre d’Atenció Primària Raval Sud, Institut Català de la Salut, 08001 Barcelona, Spain; sgonzalez.bcn.ics@gencat.cat; 16Instituto de Neuropsiquiatria y Adicciones del Parc de Salut del Mar (INAD), Consorci Parc de Salut Mar, 08001 Barcelona, Spain; LMMartin@parcdesalutmar.cat; 17Department of Psychiatry and Legal Medicine, Universitat Autònoma de Barcelona, Bellaterra, 08290 Cerdanyola del Valles, Spain

**Keywords:** depression, primary health care, chronic physical illness, nurses, psychoeducation

## Abstract

The association between physical illness and depression implies a poorer management of chronic disease and a lower response to antidepressant treatments. Our study evaluates the effectiveness of a psychoeducational group intervention led by Primary Care (PC) nurses, aimed at patients of this kind. It is a randomized, multicenter clinical trial with intervention (IG) and control groups (CG), blind response variables, and a one year follow-up. The study included 380 patients ≥50 years of age from 18 PC teams. The participants presented depression (BDI-II > 12) and a physical comorbidity: diabetes mellitus type 2, ischemic heart disease, chronic obstructive pulmonary disease, and/or asthma. The IG (n = 204) received the psychoeducational intervention (12 weekly sessions of 90 min), and the CG (n = 176) had standard care. The patients were evaluated at baseline, and at 4 and 12 months. The main outcome measures were clinical remission of depressive symptoms (BDI-II ≤ 13) and therapeutic response (reduction of depressive symptoms by 50%). Remission was not significant at four months. At 12 months it was 53.9% in the IG and 41.5% in the CG. (OR = 0.61, 95% CI, 0.49–0.76). At 4 months the response in the IG (OR = 0.59, 95% CI, 0.44–0.78) was significant, but not at 12 months. The psychoeducational group intervention led by PC nurses for individuals with depression and physical comorbidity has been shown to be effective for remission at long-term and for therapeutic response at short-term.

## 1. Introduction

Depression affects around 300 million individuals worldwide, that is to say 4.4% of the population, and represents a frequent cause of morbidity, disability, and loss of productivity [1,2]. The World Health Organization (WHO) estimates that depression will be the main health problem by 2030 [3]. This disorder, which is common in Primary Care (PC) consultations, represents one of the main reasons for the loss of quality-adjusted life years (QALYs) [4]. There is some evidence linking physical illness and depression. According to the National Institute for Health and Care Excellence (NICE) [5] guidelines, depression is present in 20% of individuals presenting a chronic physical pathology, and its correct treatment may improve both life expectancy and its quality. Numerous studies have reported that in individuals with ischemic heart disease (IHD) the prevalence of postinfarction depression is around 27% which is considerably higher than that observed in the general population [6,7,8,9]. In the case of type 2 diabetes mellitus (DM), depression is associated with a worse quality of life, lower adhesion to pharmacological treatment, greater metabolic decompensation, and decreased preventive self-care irrespective of the individual’s chronic pathology [10,11,12]. Other studies with patients suffering from chronic obstructive pulmonary disease (COPD) have observed depression to be an indicator for emergency room care use and hospitalization due to acute exacerbation of the condition, irrespective of its severity. In addition, they have reported worse adherence to prescribed pharmacological treatment and a deterioration in quality of life [13,14,15,16].

Most clinical guidelines and studies [6,17,18,19] indicate psychotherapy as the first line of treatment for mild and moderate depression as it is less invasive and more effective. Nevertheless, antidepressant pharmacology continues to be the most commonly prescribed remedy in spite of the possibility of unwanted secondary effects and financial expense [20]. In our environment, the yearly cost of antidepressants per patient in 2017 was €98.1, which represented an increase of 2.9% with respect to the 2016 [21]. It should be noted that when patients who have not clinically improved under antidepressant treatment receive some type of psychotherapy, they present a better symptomatology and remission rate at short-term [16]. Patients with chronic physical comorbidities are accustomed to receiving complex treatments for the management of their pathology. Nevertheless, adding a new drug for depression management should be carefully considered in order to avoid possible interactions. Training programs can enable PC professionals detect and efficiently deal with individuals presenting both depression and physical comorbidity and thus optimize their care. The various non-pharmacological interventions that have shown to be effective in the treatment of depression include group psychoeducation [22,23,24], cognitive behavioral therapy (CBT) either in groups or individually [25,26], problem-solving therapy (PST) [27], and physical exercise, for instance, Tai-Chi [28] and indoor rock-climbing [29]. A systematic revision which assessed the efficacy of treatments for depression in adults who had experienced an acute coronary syndrome found that CBT, together with oral antidepressants, improved depressive symptomatology [6]. Other articles have described how CBT and PST improved depressive symptoms and blood glucose levels in patients with DM [30,31,32,33].

This article evaluated the effectiveness of a psychoeducational group intervention carried out by PC nurses. They had previously received training related to the objective of the study. Research, such as the study performed by Casañas et al. [23,34], has demonstrated good results for individuals with depression. We included patients with both depression and physical comorbidity. The principal objective was to evaluate the effectiveness of a group intervention through the remission rate of depressive symptoms and the therapeutic response (50% reduction in depressive symptoms with respect to initial evaluation). A secondary objective was to assess whether the improvement in depressive symptoms was related to an enhanced quality of life following the intervention and during follow-up.

## 2. Materials and Methods

### 2.1. Design and Procedure

A multi-center, randomized, clinical trial composed of an intervention group (IG) and a control one (CG). Variable responses were blinded and there was a one-year follow-up. The study methodology has been previously published [35]. The research was carried out in the PC centers, and 18 teams from urban and rural areas in Catalunya (Spain) took part. Patients were consecutively recruited by either the physician or nurse from the Primary Care Team (PCT) to which they corresponded. Informed consent was signed by the patients who were allocated by their PCT to either an IG or CG. Assignment to the groups was randomly generated by a centralized computer platform and the investigators were blinded to the outcome. Data were gathered at commencement of the study, and at four and twelve months following baseline evaluation. Patient assessment was carried out by external examiners who were unaware of which group they belonged to.

The study was approved by the Ethics Research Committee at the IDIAP Jordi Gol University Research Institute (P16/184).

### 2.2. Subjects

Taking into account a previous psychoeducational group study [23] which reported at nine-months follow-up a remission rate of depressive symptoms of 40% and 26% for the IG and CG, respectively, it was calculated that a sample of 504 individuals was required. Assuming a confidence interval (CI) of 95% and 5% precision, and anticipating 25% dropout, 252 participants were assigned to each group.

Inclusion criteria were (a) male and female patients aged >50 years; (b) presenting some of these physical comorbidities: DM, IHD, COPD, and/or asthma registered in the clinical records; (c) a rating >12 in the Beck Depression Inventory (BDI-II) according to the Spanish version adapted by Sanz et al. [36]; and (d) to have had one year follow-up in the same PCT.

Exclusion criteria were (a) patients diagnosed with dementia or moderate/elevated cognitive deterioration (5 or more errors on the Pfeiffer Scale); (b) major depressive disorder with psychotic symptoms or any other severe psychiatric comorbidities; (c) moderate/elevated risk of suicide (a rating of 6 on the MINI scale); (d) drug abuse or dependence (including alcohol); (e) physical illness at an advanced stage; (f) inability to attend the intervention; (g) referred to a psychologist at the Mental Health Community Team; and h) unable to understand the language of the tests (Spanish/Catalan).

### 2.3. Evaluation Tools and Variables

#### 2.3.1. Baseline Variables: Sociodemographic and Clinical Ones

Data concerning the following sociodemographic variables were gathered: sex, age, civil status, educational level, employment situation, environment (rural/urban population >15,000 inhabitants), and socioeconomic status according to the MEDEA economic deprivation index [37] calculated for the urban population.

##### Hamilton

The Hamilton scale, with a score 0–56, was employed to measure anxiety [38]. As it quantifies intensity rather than the presence or not of anxiety it was useful to identify variations following the intervention.

##### Adjusted Morbidity Group (AMG)

The pooled adjusted morbidity (AMG) was also collected [39]. This measurement takes into account the individual’s multimorbidity according to illness typology and, in the case that it is chronic, identifies whether it affects one or more organic system. The range is from 1 to 4 according to the organs which present the pathology, there is a fifth category for when the patient presents an active neoplasm.

##### Pharmacological Prescription

The patients’ computerized clinical records were used to obtain this information. The number of prescribed antidepressants and anxiolytics was recorded.

##### Intervention Compliance

Participants’ attendance at the sessions was recorded with a register filled in by the observing nurses. 

##### Satisfaction

On finishing the last group session the participants filled in a questionnaire in which they indicated their level of satisfaction with the intervention. The questions concerned such aspects as knowledge acquired during the sessions and its application, the organization and duration of the sessions, and the objectives reached.

#### 2.3.2. Clinical Remission and/or Depression Response at Short and Long-Term (One Year Follow-Up)

The primary outcomes were the remission rate of depressive symptoms and therapeutic response (50% reduction in depressive symptoms with respect to initial evaluation), measured on finishing the intervention (4 months) and at one year follow-up. They were evaluated with the Spanish adaptation of the BDI-II [36,40] which has shown good psychometric characteristics with respect to construct validity, internal consistence of the items, and predictive value. The questionnaire has a score ranging from 0 to 63 points. The accepted cut-off points to determine the gravity of the symptoms are: minimum depression (0–13 points), slight (14–19 points), moderate (20–28 points), and severe (29–63 points). A score of ≤13 in the BDI-II was considered remission, and therapeutic response a 50% reduction in depressive symptoms with respect to the initial evaluation.

#### 2.3.3. Secondary Outcomes: Quality of Life at Short and Long-Term (One Year Follow-Up)

Quality of life was measured by the EuroQol scale (EQ-5D) [41], the Spanish adaptation of which has shown good construct validity. It is a self-administered instrument composed of two parts, the descriptive EQ-5D system and the visual analogue scale (EQ-VAS). The former evaluates five health dimensions: mobility, personal care, daily activities, pain, and anxiety/depression. There are five response options for each item which range from “no problems” to “extreme problems”. The latter measures the individual’s global health level with a score from 0 (the worst state of health) to 100 (the best possible).

### 2.4. Psychoeducational Group Intervention

Patients assigned to the IG received a psychoeducational group intervention directed by PC nurses. The intervention consisted of 12 weekly sessions held consecutively for 3 months. Each session lasted 90 min. The program of the sessions is explained in the protocol [35]. It provided: (1) health education about chronic pathologies and depressive symptoms; (2) information regarding the relationship between a chronic pathology and depressive symptoms; (3) health education about: diet, physical exercise, sleep, agreeable activities, social skills, pharmacological treatment and adherence to it; (4) breathing techniques; (5) problem-solving techniques, behavioral activation, and cognitive-behavioral perspective about depression; and (6) confidence and assertiveness. After each session, in order to improve proactivity, the patients were assigned tasks to be carried out during the week which would be explained to the group in the following session. At the end of the final session the participants could fill out a satisfaction survey in order to evaluate the intervention. 

The sessions were conducted by two PC nurses from each participating center and held in the same location, space and equipment permitting. The nurses had received prior training based on the psychoeducational group intervention protocol [42]. The twenty-hour program was given by psychologists, community nurses, and mental health specialists who had previous experience of leading groups. It included the content of the twelve sessions in order to ensure homogenization of the intervention.

Six months after commencement of the intervention the IG patients received a telephone call from the evaluators. They were asked about their mood and general health, and whether they were better/the same/worse than when the intervention had finished. Questions were asked about some of the goals achieved related to the proposed changes in diet, physical activity, rest, sleep, and free time/entertainment. In addition, patients’ responses to whether or not they had put into practice the techniques acquired during the intervention (controlled breathing, problem-solving, assertive communication, and restructuring of thoughts) were noted. They were also questioned about any pharmacological treatment they took and their compliance. 

The CG patients received their usual attention from the PC team. They were seen by their physician/nurse without following any previously established pattern. Appointments were made according to the habitual clinical criteria corresponding to the patients’ chronic pathology or depressive symptomatology.

### 2.5. Data Analysis

The analyses were performed on an intention-to-treat basis. Missing data were accounted for through Predictive Mean Matching with ten imputations each of which has five interactions. The evaluation of the parameters of each imputation was carried out according to Rubin’s rules [43].

Descriptive statistics were employed to summarize data at a general level and for groups (IG and CG). Continuous variables are expressed by means and standard deviation (SD), the categorical categories are shown as frequencies and percentages.

So as to assess differences between characteristics (variables) according to group, chi-square test was used for categorical variables and Student’s T test for continuous ones. In both cases, a *p*-value < alpha 0.05 was considered statistically significant.

Logistic regression was employed for the main objective in order to evaluate the differences in both groups for clinical remission rate and 50% response post-intervention (4 months) and 12 months from baseline. Results are presented as crude and adjusted for sex, age, and anxiety symptoms. The Odds Ratios (OR) are shown with their 95% CI.

Finally, in order to compare the mean differences between the groups with respect to the BDI-II and Euroqol multi-level models with mixed effects were applied. In this way, the mean differences between groups were assessed and the 95% CI calculated. The statistical analyses were performed with R Software for Windows version 3.6.1, Vienna, Austria.

## 3. Results

A total of 380 patients were included with a mean age of 68.4 years (SD:8.8), women represented 81.8%. The flow of participants is depicted in Figure 1. They were randomized into two groups: the IG (n = 204) who received the psychoeducational intervention, and the CG (n = 176) who had the usual clinical attention from their PC physician/nurse. On termination of the intervention 66 participants (29 IG and 37 GC) were not evaluated as they had not answered the questionnaire. At one-year follow-up the BDI-II questionnaire had not been completed by 75 patients (IG n = 36, CG n = 39) which represents a drop-out rate of 17.6% and 22.2%, respectively. The principal causes were inability to get in touch with the patient at the time of the questionnaire, the patient declining to continue in the study, the patient located out of the area or hospitalized, and, in the case of two participants, death (Figure 1). The mean time from the first evaluation was 4.4 months (SD 1.3) for the post-intervention, and 11.5 months (SD 0.7) for the final evaluation for both groups. 

Table 1 shows the baseline characteristics for both the IG and CG subjects. The groups were homogenous with respect to sociodemographic and clinical variables with the exception of asthma which was greater in the IG (*p* = 0.030). The most common sociodemographic profile was that of a woman, aged between 50 and 69 years, urban-dwelling, married, and retired who had finished primary education and presented high/medium-high socio-economic hardship. She suffered from chronic illnesses in more than four systems or organs (the most prevalent being DM), presented mobility problems and pain according to the eEQ-5D, and was prescribed antidepressants. 

Remission and therapeutic response results with imputed data are shown in Table 2. Remission represented 53.9% and 41.5% in the IG and GC, respectively, at twelve months of the intervention. A finding that was statistically significant (OR = 0.61, 95%CI, 0.49–0.76). These values were replicated when adjusted for age and sex (OR = 0.61, 95% CI, 0.48–0.76), and for age, sex, and anxiety according to Hamilton (OR = 0.61, 95%CI, 0.49–0.76). The therapeutic response improved at four months post-intervention in the IG (OR = 0.59, 95%CI, 0.44–0.78). When the model was adjusted for age and sex (OR = 0.59, 95%CI, 0.44–0.78), and age, sex, and anxiety according to Hamilton (OR = 0.59, 95% CI, 0.44–0.78) it did not vary. 

We also observed findings related to remission and therapeutic response in the patient subgroups according to the degree of depression (slight or moderate) in both IG and CG (Table 2). With respect to the IG, in patients with moderate depression, remission was observed at 12 months post-intervention (OR = 0.50, 95%CI, 0.35–0.71), and therapeutic response at 4 months (OR = 0.50, 95%CI, 0.35–0.71) and 12 months (OR = 0.53, 95% IC, 0.38–0.76). The values were maintained when the data were adjusted in the different models. The participants with slight depression in the IG had a remission at 12 months (OR = 0.66, 95%CI, 0.49–0.90), and a therapeutic response that was only statistically significant at 4 months (OR = 0.66, 95%CI, 0.45–0.97). 

The therapeutic response was greater in the IG than in the CG at both 4 and 12 months according to the mean BDI-II values without achieving statistical significance (Table 3). With respect to the EQ-VAS, whilst the IG quality of life at 4 months was better perceived, at 12 months it matched that of the CG (Table 3). 

Table 4 depicts the EQ-5D dimensions according to subgroups. Whilst the percentage of subjects who presented problems in the various dimensions was always higher in the CG than the IG at 4 and 12 months (with the exception of pain), these differences were not statistically significant. Nevertheless, the increase in the number of participants who had problems at 12 months in the CG was greater than in the IG with respect to mobility, selfcare, daily activities, and pain. In the area of depression and anxiety the percentages of subjects who presented problems was greater in the CG at 4 months (IG = 82.3% CG = 87.8%), and the same in both groups at 12 months (IG = 78.6 CG = 78.8).

At the end of the evaluation 61.9% and 62.0% of the IG and CG, respectively, were taking antidepressants. The percentage of subjects receiving these drugs remained the same in the IG (61.3% at commencement vs. 61.9% at 12 months), and increased in the CG (59.1 at commencement vs. 62.0%), without being statistically significant.

Mean attendance at the sessions was 7.2 (SD 4.24; range 0–12) and 65.2% of the participants (n = 133) received 7 or more sessions (Table 5). At 12 months those subjects that presented a remission had attended a mean of 8.5 sessions (SD 3.7), and those who did not present a depression remission had attended a mean of 7.1 (SD 4.2), with a statistical significance between groups (*p* = 0.021). Data concerning the satisfaction questionnaire are shown in Table 5. The level of satisfaction was a mean of 9.3 (SD 1.2) with a maximum of 10 and a minimum of 3. The topics that were considered the most useful were, in the following order: 1. Cognitive restructuring; 2. Behavioral activation and the education and identification of symptoms; 3. Diet; and 4. Breathing and relaxation techniques. Those considered less useful were: physical activity, use of time, and health education about pharmacological treatment. 

## 4. Discussion

Depression remission in patients presenting both depression and comorbidity was 12% higher in those who received the psychoeducational group intervention led by PC nurses than the control. The effect of the intervention on the response to depressive symptomatology was greater at short-term: a positive response in the IG was observed at 4-months post-intervention but not at 12. Such findings indicate that the intervention is efficient at short-term in decreasing symptomatology and at long-term in depression remission. In our sample more than 40% were aged over 70 years. In this regard, Wilkinson et al. [44] suggested that elderly individuals take longer to adapt to change. A possibility that could explain why our participants might have needed more time to assimilate the knowledge and skills imparted during the intervention sessions, and the reason remission appeared at long-term.

For the two IG depression severity subgroups (slight and moderate), a response at 4 months and remission at 12 months were greater, with statistical significance for both. Nevertheless, in the moderate depression subgroup the 50% improvement in symptomatology was maintained long-term up to 12 months, but not in the slight depression one. Such minor differences signify that intervention should not be focused solely on patients with moderate depression. Studies have demonstrated the efficacy, in a preventive manner, of behavioral activation and CBT in subjects with slight depression, thus avoiding the development of more severe symptoms [45]. 

Our results regarding effectiveness are similar to those of Casañas et al. [23,34,46] who also reported remission in the IG in a program very similar to ours, carried out by PC nurses but focused on subjects with depression from the general population. A systematic revision carried out by Kastner et al. reported that collaborative interventions with different care levels have been shown to be effective in improving depressive symptoms in adults with DM and cardiovascular disease [47]. Other studies have incorporated combinations of CBT and physical exercise into community-based interventions. Positive results were obtained with a 71% depression remission in patients with DM [32]. In addition, a study also performed in PC with CBT, but by final year medical students previously trained by psychiatrists, aimed at patients with depression and DM, obtained a significant decrease in the BDI-II [31]. 

Our intervention included psychoeducation related to depression and health education for the various comorbidities the patients presented. Emphasis was placed on healthy lifestyle habits: diet, physical activity, and rest. Such a mixed focus has been shown to be effective in other studies, mainly in patients with DM [32,33]. In contrast, other programs led by nurses for patients with DM and IHD, which only included CBT, did not obtain significant results (OR = 1.21, CI 95% 0.12 to 12.41) with respect to depressive symptoms [48]. 

Regarding our second objective, quality of life did not improve significantly with depression remission. Whilst an increase in the global ratings of the EQ-VAS in the IG was observed after the intervention, the findings were not statistically significant when compared to those in the CG. As reported by Orfila et al., quality of life in elderly individuals appears to strongly related to disability and functional limitation [49]. This could explain why it is not significantly affected by interventions aimed at improving depression. Indeed, other PC programs with patients presenting a similar profile to ours, aimed at promoting support and social participation in elderly individuals with a bad self-perception of health, did not report improvements in the subjects’ quality of life [50]. 

Research such as ours which evaluates a psychoeducational group intervention carried out by PC nurses for patients presenting depression and chronic organic pathology is scarce. The community nurse is the PC professional who is generally responsible for the follow-up of patients with chronic pathologies. It is, therefore, clear that training these nurses in group interventions is the most efficient way to deal with depression as one more comorbidity in such patients. The capacity of PC nurses, who have received prior training in mental health, to identify, evaluate, and manage the mental health risk of patients with chronic pathologies within a collaborative care regime has been demonstrated [51]. Aragonés et al. [52] reported that when nurses in charge of patient clinical follow-up encouraged treatment adherence and provided psychoeducational care to IG patients and their families, this resulted in favorable rates of remission (48.8% IG vs. 35.6% CG *p* = 0.026) and response to depression (66% IG vs. 51% CG *p* = 0.011). Qualitative research has evaluated impressions from both patients with depression and physical comorbidity (for instance, DM and IHD) and the PC nurses who attended them. It identified a low perception of need for care related to depression from the nurses, and the difficulty for patients to actively ask for help [53]. Evaluating the patient’s state of mind should form part of habitual clinical practice. The trust established between patients with chronic pathologies and their professionals of reference (doctors and nurses) should enable them to express their emotions, and permit negative states of mind such as depression be identified. Patients need time to speak and reflect on their feelings; their refusal to ask for help could be due to the stigma and negative connotations that depression continues to suffer from in our society. Lack of awareness about this condition, and the limitations some nurses may experience in addressing emotional distress, are barriers that could be overcome by our intervention. This could result in a correct management of patients similar to our study participants.

The group format of our intervention is an added value. It is well-known that group interventions provide therapeutic advantages and factors that differ from individual interventions [54]. Such benefits include social support, the feeling of belonging to a group, and the experience of sharing common problems.

The dropout rate in our study was lower than others carried out with patients presenting similar characteristics [23,50]. Attendance at the sessions was, however, moderate with a mean of seven sessions which has led to the possibility of shortening them. Such a figure is a little lower than that obtained by Casañas et al. who reported a mean of nine sessions [23], and in which there was also a decrease of the BDI-II in the IG. It is possible that the health status of the participants with comorbidities played a part. Such patients often feel unwell which hinders their leaving home to attend appointments. The support telephone call at six months probably influenced the good results obtained at long-term. Experience regarding telephoning in managing depression with physical comorbidities has been very positively rated by participants [55,56]. Such interventions could be a viable alternative when moving to a health center is complex, or in contexts such as the current Covid-19 pandemic where gatherings of patients represent a risk of infection. Telematically adapting interventions such as ours to maintain efficient management of depression, and its associated physical pathologies, could be an interesting possibility for the future. A meta-analysis assessing CBT efficacy through guided and non-guided line evaluation with respect to other kinds of interventions to treat depression, reported mean adherence rates very similar to ours, 76% in the case of guided CBT [57]. The therapeutical response of the patients in the guided line intervention was 48%, a figure close to the one we obtained at 12 months.

Consumption of medication was not reduced in the IG in spite of depression remission, and slightly increased in the CG. Such a finding may reflect the predisposition PC physicians have to introduce antidepressant treatment, and their reluctance to withdraw it at long-term despite the patient’s improvement. Such preferences do not always coincide with those of the patients who have been shown to be more in favor of some kind psychotherapy to treat depression rather than pharmacological treatment [58]. Moreover, as the current pandemic has resulted in a considerable increase in healthcare costs, interventions such as ours could have a better cost-effectiveness ratio.

One of the strengths of our study was that the intervention was carried out in an environment the patients frequently visited. In addition, the profile of the participants corresponded to that of the patients who generally attended PC consultations. Another strength is the study’s design, it was a randomized clinical trial with homogeneity between the IG and CG, and with a highly representative sample from both urban and rural areas. As the randomization was blinded the evaluations and statistical analysis provided greater validity to the results.

With regard to limitations, one of these is the lack of data in some subgroups with respect to civil status and employment situation which prevented us from carrying out an analysis from a social perspective. Another is that we did not make any distinctions regarding the date of diagnosis of depression. This meant we did not know whether the diagnosis of comorbidity was prior to diagnosis of depression or afterwards. Moreover, the fatigue resulting from dealing with chronic pathologies could be confused with a depressive state. Neither was the time of the evolution of depression assessed, another key factor in resistance to treatment. Another factor to take into consideration is the low participation of men which reflects the gender gap [59] regarding susceptibility to depression. Finally, at four months 21% of the IG participants and 8% of the CG had told the evaluator which group they belonged to. At 12 months the values were 23% and 8% for the IG and CG, respectively.

## 5. Conclusions

The psychoeducational group intervention conducted by PC nurses for patients with depression and physical comorbidity proved effective for depression remission at long-term and for therapeutic response short-term. Quality of life did not change significantly in both groups following the intervention.

## Figures and Tables

**Figure 1 ijerph-18-02948-f001:**
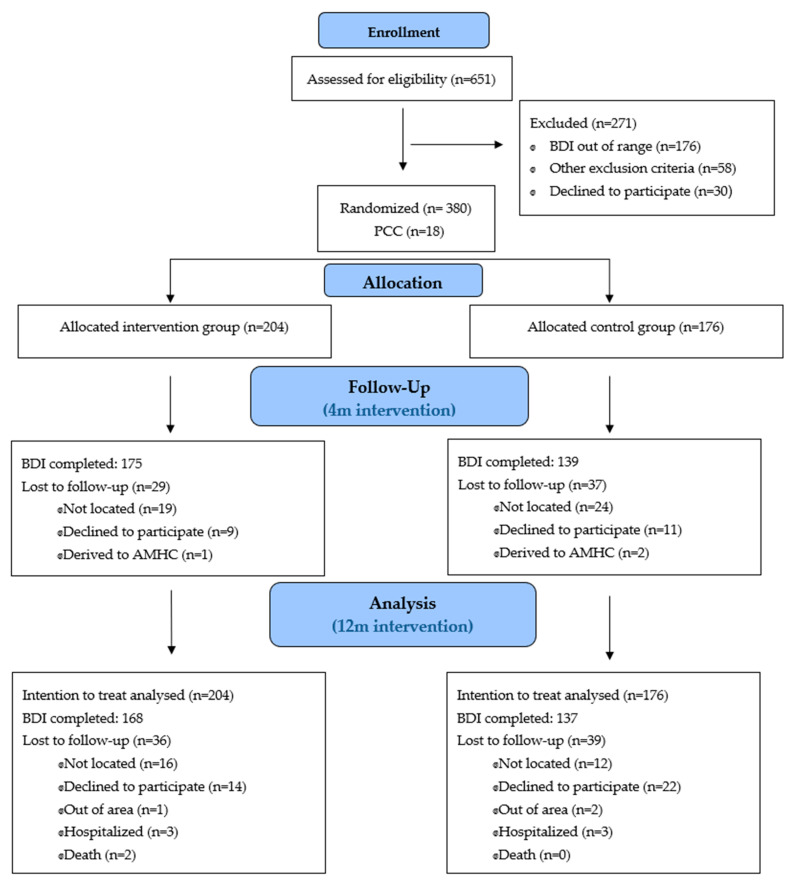
The flow of participants. Abbreviations: PCC, primary care center. AMHC, Adult mental health center. BDI, Beck depression inventory.

**Table 1 ijerph-18-02948-t001:** Sociodemographic and clinical variables at baseline.

	GLOBAL		INTERVENTION (IG)		CONTROL (CG)		
	N = 380		N = 204		N = 176		
**Sociodemographic variables**	n	%	n	%	n	%	*p*
**Gender**							0.581
Men	70	18.4	35	17.2	35	19.9	
Women	310	81.6	169	82.8	141	80.1	
**Age**							0.150
From 50 to 69 years	204	53.7	117	57.4	87	49.4	
>70 years	176	46.3	87	42.6	89	50.6	
Environment							0.920
Rural	84	22.1	46	22.5	38	21.6	
Urban	296	77.9	158	77.5	138	78.4	
**Socioeconomic deprivation (MEDEA index *)**							0.446
Low/medium low deprivation	141	47.6	72	45.6	69	50.0	
High/medium high deprivation	155	52.4	86	54.4	69	50.0	
**Educational level**							0.461
No studies/incomplete primary studies	97	25.7	47	23.2	50	28.7	
Completed primary education	159	42.2	88	43.3	71	40.8	
Secondary education/University	121	32.1	68	33.5	53	30.5	
Missing	3	0.80	1	0.5	2	1.1	
**Civil status**							0.283
Married/cohabitant	205	54.1	118	57.8	87	49.7	
Widow/widowed	100	26.4	49	24.1	51	29.1	
Divorced/separated/single	74	19.5	37	18.1	37	21.2	
Missing	1	0.3	0	0.0	1	0.3	
**Employment status**							0.683
Housewife	40	10.5	25	12.3	15	8.5	
Unemployed	30	7.9	15	7.2	15	8.5	
Incapacitated	37	9.7	22	10.8	15	8.5	
Retired	235	61.9	123	60.4	112	63.7	
Employed	38	10.0	19	9.3	19	10.8	
**Clinical variables**							
**N° of chronic pathologies included in the study**							0.387
1 pathology	307	80.8	161	78.9	146	83.0	
2 pathologies or more	73	19.2	43	21.1	30	17.0	
**Type chronic pathology ****							
Diabetes mellitus	252	66.3	134	65.7	118	67.0	0.864
Ischemic heart disease	51	13.4	25	12.3	26	14.8	0.571
Chronic obstructive pulmonary disease	60	15.8	28	13.7	32	18.2	0.295
Asthma	100	26.3	63	30.9	37	21.0	0.039
**AMG**							0.776
Chronic pathology 2 or 3 systems/organs	24	6.6	14	7.2	10	5.9	
Chronic pathology more 4 systems/organs	326	89.3	174	89.2	152	89.4	
Active neoplasia	15	4.1	7	3.6	8	4.7	
Missing	15	3.9	9	4.4	6	3.4	
**Severity of depression (BDI-II)**							0.743
Mild depression (BDI-II <= 19)	206	54.2	110	53.9	96	54.5	
Moderate depression (BDI-II >= 20)	174	45.8	94	46.1	80	45.5	
**Euroqol by dimensions**							
With mobility problems	212	55.8	118	57.8	94	53.4	0.445
With self-care problems	79	20.8	40	19.6	39	22.2	0.628
With problems in daily activities	163	42.9	91	44.6	72	40.9	0.534
With pain problems	304	80,0	162	79.4	142	80.7	0.857
With anxiety/depression problems	347	91.3	188	92.2	159	90.3	0.531
**Hamilton anxiety**	370	12.8	176	12.7	204	13.0	0.717
Missing	10	2.6	7	3.4	3	1.7	
**Prescribed medication**							
Antidepressant	229	60.3	125	61.3	104	59.1	0.742
Anxiolytics/hypnotics	183	48.2	98	48.0	85	48.3	1

* The MEDEA socio-economic index is not applicable to rural areas. Fragmented index data appears in urban areas. ** It must be considered that some patients may present several of the referenced pathologies. Abbreviations: IG, intervention group; CG, control group; BDI-II, Beck Depression Inventory; AMG, adjusted morbidity group.

**Table 2 ijerph-18-02948-t002:** Remission and therapeutic response of depression in the overall. mild and moderate depression patients. OR adjusted according to different models.

		Intervention (IG)	Control (CG)			
	Follow-Up	n = 204	n = 176	OR* (CI 95%)	ORadj** (CI 95%)	ORadj*** (CI 95%)
Remission	Months	n (%)	n (%)			
Overall	4 m	105 (51.5%)	85 (48.3%)	0.92 (0.73 to 1.17)	0.91 (0.72 to 1.16)	0.93 (0.72 to 1.18)
	12 m	110 (53.9%)	73 (41.5%)	0.61 (0.49 to 0.76)	0.61 (0.48 to 0.76)	0.61 (0.49 to 0.76)
Mild Dep.	4 m	70 (63.6%)	63 (65.6%)	1.05 (0.77 to 1.42)	1.09 (0.80 to 1.48)	1.09 (0.80 to 1.49)
	12 m	73 (66.4%)	52 (54.2%)	0.66 (0.49 to 0.90)	0.66 (0.48 to 0.91)	0.66 (0.49 to 0.91)
Moderate Dep.	4 m	35 (37.2%)	22 (27.5%)	0.76 (0.52 to 1.11)	0.77 (0.53 to 1.13)	0.81 (0.55 to 1.19)
	12 m	37 (39.4%)	21 (26.2%)	0.50 (0.35 to 0.71)	0.50 (0.35 to 0.71)	0.51 (0.36 to 0.73)
**Therapeutic response**						
Overall	4 m	64 (31.4%)	38 (21.6%)	0.59 (0.44 to 0.78)	0.59 (0.44 to 0.78)	0.59 (0.44 to 0.79)
	12 m	86 (42.2%)	69 (39.2%)	0.82 (0.65 to 1.04)	0.82 (0.65 to 1.04)	0.82 (0.65 to 1.04)
Mild Dep.	4 m	34 (30.9%)	23 (24.0%)	0.66 (0.45 to 0.97)	0.67 (0.46 to 0.98)	0.67 (0.45 to 0.98)
	12 m	42 (38.2%)	41 (42.7%)	1.22 (0.87 to 1.70)	1.20 (0.86 to 1.68)	1.21 (0.86 to 1.70)
Moderate Dep.	4 m	30 (31.9%)	15 (18.8%)	0.51 (0.35 to 0.75)	0.52 (0.36 to 0.77)	0.54 (0.37 to 0.79)
	12 m	44 (46.8%)	28 (35.0%)	0.53 (0.38 to 0.76)	0.54 (0.38 to 0.77)	0.55 (0.39 to 0.78)

OR* un adjusted model. Group variable (IG = 1/CG = 0); ORadj** model adjusted by group (IG = 1/CG = 0), sex and age. OR adj*** model adjusted by group (IG = 1/CG = 0), sex, age, and Hamilton. Depression Remission (BDI-II ≤ 13). Therapeutic response: improvement of symptoms >50% with respect to baseline. Mild Depression (BDI-II ≤ 19); Moderate Depression (BDI-II ≥ 20). Abbreviations: IG, intervention group; CG, control group; CI, confidence interval.

**Table 3 ijerph-18-02948-t003:** Mean value (SD) of BDI-II and VAS Euroqol baseline, 4 m and 12 m. Changes BDI-II and VAS euroquol between groups.

	INTERVENTION (IG)		CONTROL (CG)		Difference (95% CI) between Groups	
Follow-Up	N = 204		N = 176		(Intervention Group -Usual Care Group) **	
**Months**	mean (SD)	Difference * (SD)	mean (SD)	Difference * (SD)	Difference * (95% CI)	*p*
**BDI-II**						
Baseline	19.7 (5.2)		19.3 (5.2)			
4m	14.9 (8.2)	−4.8 (8.0)	15.4 (8.1)	−3.9 (7.5)	−1.37 (−2.22 to −0.52)	0.110
12m	15.0 (9.2)	−4.7 (8.6)	15.2 (9.3)	−4.1 (8.4)	−0.60 (−1.49 to 0.27)	0.493
**VAS euroqol**						
Baseline	55.8 (16.7)		56.6 (20.1)			
4m	61.0 (18.8)	5.2 (19.5)	59.0 (20.4)	2.4 (18.9)	2.58 (0.40 to 4.76)	0.238
12m	59.4 (20.9)	3.6 (22.4)	59.7 (21.0)	3.1 (23.0)	−0.06 (−2.75 to 2.64)	0.983

* Differences were calculated between baseline measurement and follow-up measurement. ** Differences were calculated between intervention group and control. Abbreviations: BDI-II, Beck Depression Inventory; VAS euroquol, Visual Analogical Scale; SD, Standard Derivation.

**Table 4 ijerph-18-02948-t004:** EQ-5D dimensions for intervention group and control at baseline, 4 months, and 12 months.

	GLOBAL		INTERVENTION (IG)	CONTROL (CG)			
Euroqol by Dimensions	n	%	n	%	n	%	*p*
**With mobility problems**							
Baseline	212	55.8	118	57.8	94	53.4	0.445
4 m	181	57.6	97	55.4	84	60.4	0.373
12 m	181	59.3	96	57.1	85	60.2	0.386
**With self-care problems**							
Baseline	79	20.8	40	19.6	39	22.2	0.628
4 m	89	28.3	49	28.0	40	28.8	0.879
12 m	97	31.8	49	29.2	48	35.0	0.274
**With problems in daily activities**							
Baseline	163	42.9	91	44.6	72	40.9	0.534
4 m	173	55.1	96	54.9	77	55.4	0.924
12 m	162	53.1	85	50.6	77	56.2	0.329
**With pain problems**							
Baseline	304	80.0	162	79.4	142	80.7	0.857
4 m	251	79.9	144	82.3	107	77.0	0.243
12 m	246	80.7	133	79.2	113	82.5	0.466
**With anxiety/depression problems**							
Baseline	347	91.3	188	92.2	159	90.3	0.531
4 m	266	84.7	144	82.3	122	87.8	0.180
12 m	240	78.7	132	78.6	108	78.8	0.956

Abbreviations: IG, intervention group; CG, control group; EQ-5D, Euroquol 5 dimensions.

**Table 5 ijerph-18-02948-t005:** Group intervention. Session attendance and level of satisfaction.

Attendance at group sessions (0 to 12)	n = 204
mean (SD)	7.2 (4.2)
median (interquartile range)	9.0 (3.0–11.0)
Level of satisfaction with the intervention	n = 122
mean (SD)	9.3 (1.2)
median (interquartile range)	10.0 (9.0–10.0)
Items regarding satisfaction with the intervention	Percentage YES
Has your knowledge about depression improved?	93.1
Do you think the knowledge you gained during the sessions can be applied to your daily life?	99.2
Have the topics discussed during each session been what you expected?	90.9
Have the explanations about the topics been understandable?	99.2
Have you obtained, in general, support from the group?	98.4
Have the group sessions been participative?	98.3
Have you been able to express your feelings and thoughts?	96.7
Do you think the organisation of the sessions has been adequate?	99.2
Have you achieved the expected goals of the group?	94.1
Do you think the length of each session is correct?	90.8
Do you think the duration of the group intervention has been correct?	92.6
In general, are you glad you participated in the intervention?	99.2
Would you recommend this kind of group intervention to a relative or friend?	98.3

Abbreviations: SD, Standard Derivation.

## Data Availability

All the principal investigators of the study had access to the complete database, and the datasets generated and analysed during the current study will be available from the corresponding author.

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
