# Peer review of "Effectiveness of a Psychoeducational Group Intervention Carried Out by Nurses for Patients with Depression and Physical Comorbidity in Primary Care: Randomized Clinical Trial"

_ijerph, 2021, doi:10.3390/ijerph18062948_

Round 1

Reviewer 1 Report

Thanks for allowing me to review this interesting article. In general, the paper is well-written. I have some questions though.

  1. Line 75-76, please provide reference.
  2. I have seen so many different therapies (i.e. CBT, Tai chi, etc from your article) that have been done to improve the quality of life of patients who have both chronic disease and depression. What make you think that psychoeducation should be done to this group of patient? In other word, why psychoeducation so special?
  3. Literature review in the introduction provides little information on previous work of psychoeducation program on this group of patients.
  4. Have you considered to conduct a subgroup analysis for identifying the effect of the program on patients with different chronic disease (i.e. COPD, DM...)?
  5. If word count is allowed, please indicate the validity and reliability of each instrument you use in this study.
  6. I cannot see the primary outcome of this program.
  7. Were the interventions guided by a conceptual framework? Were the interventions provided individually or in group? Why?
  8. It is not clear whether the program has used cluster randomization. It seems to me that ICC is not included in calculating the sample size.
  9. The attendance rate for the program was not high (the paper claimed that it is because of the poor health status of this group) , have you asked about the reason behind?
  10. There are many psychoeducation program that has been done to elderly people (most probably have chronic diseases), how to distinguish your study with these others?

Author Response

Dear reviewer, I proceed to respond to your comments:
  1. Line 75-76, please provide reference. The following reference has been added: Martín, J. C., Garriga, A., Egea, C., Diaz, G., Campillo, M.-J., & Espinosa, R. M. (2017). Intervención psicológica escalonada con trastornos mentales comunes en Atención Primaria. Anales De Psicología / Annals of Psychology34(1), 30-40. https://doi.org/10.6018/analesps.34.1.281491
  2. I have seen so many different therapies (i.e. CBT, Tai chi, etc from your article) that have been done to improve the quality of life of patients who have both chronic disease and depression. What make you think that psychoeducation should be done to this group of patient? In other word, why psychoeducation so special? We considered that psychoeducation for this kind of patient lay within the principles and skills of primary care nurse responsibilities. Psychoeducation aims to empower patients so as to be able to correctly cope with their health issues, and boost their autonomy by providing resources that can be used in managing both their depression and chronic pathology. Other kinds of therapy do not provide such a holistic approach to the patients’ needs.
  3. Literature review in the introduction provides little information on previous work of psychoeducation program on this group of patients. It is certain that there is little research providing results from psychoeducational interventions carried out in primary care with this kind of patient. For this reason, we were interested in performing our study.
  4. Have you considered to conduct a subgroup analysis for identifying the effect of the program on patients with different chronic disease (i.e. COPD, DM...)? This is one of our study limitations which we explain in the Discussion section of the article. Such an analysis was not considered for two reasons (see Table 1). The first was that subgroups implied very small samples in order to obtain significant results; the second was due to the fact that there were patients who presented more than one associated chronic physical pathology, they would thus simultaneously belong to more than one subgroup.
  5. If word count is allowed, please indicate the validity and reliability of each instrument you use in this study. In the Methods section we have added an explanation about the characteristics of the instruments evaluating the primary and secondary outcomes.
  6. I cannot see the primary outcome of this program. For clarification regarding which are the primary and secondary outcomes we have made some changes to the paragraph (2.3.2) in the Methodology section where this information appears.
  7. Were the interventions guided by a conceptual framework? Were the interventions provided individually or in group? Why? The conceptual framework of the intervention was from that of a community nurse. It was aimed at obtaining the greatest possible autonomy for chronic patient selfcare. As these patients spend little time in consultations, they need to be able to take responsibility for their own chronic health issues, acquire skills in managing them, and be aware of possible decompensations and how to resolve them. The intervention was a group one as stated in the Methodology section (2.4). A group format was chosen in order to obtain the corresponding therapeutic benefits, this is explained in the Discussion (lines 413-416).
  8. It is not clear whether the program has used cluster randomization. It seems to me that ICC is not included in calculating the sample size. In each primary healthcare center patients were randomly assigned to the study groups. As a result, each center had patients in an intervention group and patients in a control. For this reason, we did not consider the selection by clusters.
  9. The attendance rate for the program was not high (the paper claimed that it is because of the poor health status of this group) , have you asked about the reason behind? At the end of each session the observing nurses telephoned the absent participants and asked them why they had not attended. The responses were noted in the observation register. In most cases the participants replied that they had not felt well that day, or the day of the intervention coincided with another medical appointment or test.
  10. There are many psychoeducation program that has been done to elderly people (most probably have chronic diseases), how to distinguish your study with these others? Our study is distinct because it was carried out in a primary care environment, and directed and conducted by nurses. As primary care is the location where most patients with chronic illnesses are attended it appeared to be the most suitable. Moreover, the fact that it was the primary care nurses themselves (having previously received training) who carried out the interventions makes our study different. Many psychoeducational programs for elderly subjects with depression are performed by nurses specialized in mental health or psychologists, and almost always within a specialized environment.

Kind regards,

Antonia Raya Tena

Reviewer 2 Report

This is a randomized controlled study for investigating the short-term effects (4th and 12th months) of a non-invasive educational intervention program on depression.  The study design is properly executed and presented.  Results confirm the beneficial effect of a psychological-educational intervention program.  Although the study presents interesting results and helpful suggestions for improving the RCT on depression, there are three methodological issues pertaining the presentation should be addressed as follows:

  1. Intensity of the treatment/intervention effects:  It is unclear about the details on the dose-response relationship between the outcomes and intervention.  Because the intervention constitutes multiple components, the fidelity of the treatment effects should be described.  The intervention was executed for 12 months with a total of 20 hours.  It is imperative to describe if there is any variation in regard to the dose-effect relationship of the intervention.
  2. Short-term effect: The observation is limited to a period of 12 months with a three-wave study.  Long term benefits of the intervention can not be clearly revealed from a one-year study.    
  3. Multi-wave data analysis:  The authors should address how time-varying predictor (such as the period-specific intervention by hours) could be included in the longitudinal analysis.  For instance, the intervention (type and amount) during each wave period) could be used as a time-varying predictor of the outcome(s).

Overall, I am impressed by the paper.  With the above amendments, it will certainly enhance the quality of a RCT study on depression.

Author Response

Dear reviewer, I proceed to respond to your comments:

  1. Intensity of the treatment/intervention effects: It is unclear about the details on the dose-response relationship between the outcomes and intervention.  Because the intervention constitutes multiple components, the fidelity of the treatment effects should be described.  The intervention was executed for 12 months with a total of 20 hours.  It is imperative to describe if there is any variation in regard to the dose-effect relationship of the intervention.

The intervention consisted of 12 weekly sessions, each session lasting 90 minutes. It was carried out consecutively, that is to say, over 3 months. What we have been able to associate with the primary outcome is adherence to the group intervention sessions. In the Methods and Results sections we have further clarified and amplified this information.

  1. Short-term effect: The observation is limited to a period of 12 months with a three-wave study. Long term benefits of the intervention can not be clearly revealed from a one-year study.

We are extremely grateful for this observation and are considering a longer term follow-up. We believe that evaluating this kind of patient at 12 months after intervention commencement was important in order to obtain data about its effectiveness. Maintaining the same treatment for 12 or more months, without apparent changes, justifies exploring another kind of approach.

  1. Multi-wave data analysis: The authors should address how time-varying predictor (such as the period-specific intervention by hours) could be included in the longitudinal analysis. For instance, the intervention (type and amount) during each wave period) could be used as a time-varying predictor of the outcome(s).

We measured intervention adherence by the number of sessions attended not hours. Each session lasted 90 minutes. The sessions were consecutively held every week for 3 months. At the end of the 3 months the 12 weeks of the intervention had been completed. An analysis regarding depression remission (the main result of the study) at 12 months and session attendance was carried out. This is reported in the Results section (lines 331-333).

Kind regards,

Antonia Raya Tena

Round 2

Reviewer 2 Report

The explanation of the experimentation and assessment of the intervention program is acceptable.  No other concerns are noted.